# (+)-Bornyl *p*-Coumarate Extracted from Stem of *Piper betle* Induced Apoptosis and Autophagy in Melanoma Cells

**DOI:** 10.3390/ijms21103737

**Published:** 2020-05-25

**Authors:** Yu-Jen Wu, Tzu-Rong Su, Chi-I Chang, Chiy-Rong Chen, Kuo-Feng Hung, Cheng Liu

**Affiliations:** 1Department of Beauty Science, Meiho University, Pingtung 91202, Taiwan; x00002180@meiho.edu.tw; 2Department of Food and Nutrition, Meiho University, Pingtung 91202, Taiwan; 3Antai Medical Care Corporation, Antai Tian-Sheng Memorial Hospital, Pingtung 92842, Taiwan; a081002@mail.tsmh.org.tw; 4Department of Biological Science and Technology, National Pingtung University of Science and Technology, Pingtung 91201, Taiwan; changchii@mail.npust.edu.tw; 5Department of Life Science, National Taitung University, Taitung 95002, Taiwan; gina77@nttu.edu.tw; 6Yu Jun Biotechnology Co., Ltd., Kaoshiung 91363, Taiwan; fwind101@gmail.com; 7Department of Dental Technology, Shu-Zen Junior College of Medicine and Management, Kaoshiung 82144, Taiwan; 8Department of Health Beauty, Shu-Zen Junior College of Medicine and Management, Kaoshiung 82144, Taiwan

**Keywords:** melanoma, (+)-bornyl *p*-coumarate, apoptosis, autophagy

## Abstract

(+)-Bornyl *p*-coumarate is an active substance that is abundant in the *Piper betle* stem and has been shown to possess bioactivity against bacteria and a strong antioxidative effect. In the current study, we examined the actions of (+)-bornyl *p*-coumarate against A2058 and A375 melanoma cells. The inhibition effects of (+)-bornyl *p*-coumarate on these cell lines were assessed by 3-(4,5-Dimethylthiazol-2-yl)-2,5- diphenyltetrazolium bromide (MTT) assay and the underlying mechanisms were identified by immunostaining, flow cytometry and western blotting of proteins associated with apoptosis and autophagy. Our results demonstrated that (+)-bornyl *p*-coumarate inhibited melanoma cell proliferation and caused loss of mitochondrial membrane potential, demonstrating treatment induced apoptosis. In addition, western blotting revealed that the process is mediated by caspase-dependent pathways, release of cytochrome *C*, activation of pro-apoptotic proteins (Bax, Bad and caspase-3/-9) and suppression of anti-apoptotic proteins (Bcl-2, Bcl-xl and Mcl-1). Also, the upregulated expressions of *p*-PERK, p-eIF2α, ATF4 and CCAAT/enhancer-binding protein (C/EBP)-homologous protein (CHOP) after treatment indicated that (+)-bornyl *p*-coumarate caused apoptosis via endoplasmic reticulum (ER) stress. Moreover, increased expressions of beclin-1, Atg3, Atg5, p62, LC3-I and LC3-II proteins and suppression by autophagic inhibitor 3-methyladenine (3-MA), indicated that (+)-bornyl *p*-coumarate triggered autophagy in the melanoma cells. In conclusion, our findings demonstrated that (+)-bornyl *p*-coumarate suppressed human melanoma cell growth and should be further investigated with regards to its potential use as a chemotherapy drug for the treatment of human melanoma.

## 1. Introduction

Melanoma is an invasive malignancy caused by uncontrolled division of melanocytes and responds poorly to conventional cancer treatments such as chemotherapy and radiation therapy. It is the cause of most skin cancer deaths [1,2] and the global incidence of melanoma has increased significantly, heightening the interest of researchers in Europe and North America [3,4,5]. According to the American Cancer Society, more than 91,000 new cases of melanoma were diagnosed in the United States in 2018 [4]. Melanoma can be treated with surgical resection if found early; however, deep melanoma tends to metastasize and spread to other parts of the body [6,7], rendering surgery insufficient in the later stages and making treatment more difficult. The only treatment options that remain are chemotherapy, radiation therapy and immunotherapy [8].

Chemotherapy drugs inhibit cancer proliferation or directly destroy cancer cells in cancer treatment, processes that often promote pathways that trigger apoptosis in cancer cells, leading to cell death. This approach is a widely-used method that can effectively treat cancer. The intrinsic and extrinsic pathways are the two main routes by which apoptosis is induced in cells. Loss of mitochondrial membrane potential (ΔΨm) and extensive endoplasmic reticulum (ER) stress are important characteristics of the intrinsic pathway [9], in which Bcl-2 family proteins, such as Bax, play crucial roles in collapsing the ΔΨm and causing the release of other factors, initiating apoptosis. Additionally, translocation of Bax from the cytosol to the outer mitochondrial membrane during apoptosis allows cytochrome *C* release, consequently activating caspase-9 and downstream caspase-3, which then leads to activation of poly(ADP-ribose) polymerase-1 (PARP-1) [10,11]. The ER has vital functions in many cellular responses, including in quality control of protein folding, regulation of protein synthesis and perturbation of calcium homeostasis. Accumulation of massive amounts of unfolded proteins in the ER impairs the homeostasis of the ER, resulting in stress and activating apoptosis.

Autophagy is one of the major degradation systems in eukaryotic cells, confirming that a stable balance exists between anabolism and catabolism. It is known that autophagy plays important roles in two distinctive systems, both promoting and preventing cell death. Autophagy promotes tumor suppression by eliminating severely impaired organelles and misfolded protein aggregates [12]; in addition, defects in autophagy are associated with cancer and aging [13,14]. Autophagy also triggers a non-apoptotic cell death pathway that slows down tumorigenesis, thereby obstructing tumor development [15,16]. The active role of autophagy in tumors has rendered it a potential target for development as a therapeutic strategy. Obtaining a deep understanding of the relationship between autophagy and tumor development is key to unlocking this treatment approach.

*Piper betle* Linn. belongs to the Piperaceae family and is known as the betel vine. It is a perennial, semi-woody climber with a distinct aromatic odor and a sharp bitter taste that is found in many regions of Southeast Asian countries. In some areas, such as Taiwan and India, *P. betle* is an important medicinal and economic plant that has been proven to exert antioxidative, antimicrobial and anti-hemolytic activities [17,18,19]. High contents of diverse bioactive constituents including lignins, polyphenols, alkaloids, steroids, saponins, tannins, flavonoids and terpenes have been found in the leaves and stem of *P. betle* [20]. (+)-Bornyl *p*-coumarate is a newly-identified active substance that can be obtained from the stem of *P. betle*. In this current investigation, the effect of (+)-bornyl *p*-coumarate on the cell viability of melanoma cells was evaluated using flow cytometric analysis, immunofluorescence staining and immunostaining.

## 2. Results

### 2.1. Characterization of the Constituents of the EA Fraction of the Piper Betle Stem 

A pure compound was afforded from the ethyl acetate (EtOAc) fraction of the *P. betle* stem and was identified as (+)-bornyl *p*-coumarate by comparison of physical and spectral data (specific rotation, Mass spectrometry and NMR) with the values described in the literature. (+)-Bornyl *p*-coumarate: amorphous white powder; [α${]}_{D}^{27}$ = +14.2° (*c* 0.4, CHCl_3_); ^1^H-NMR (400 MHz, CDCl_3_) δ 0.86 (3H, s, H-10’), 0.87 (3H, s, H-8’), 0.90 (3H, s, H-9’), 1.05 (1H, dd, J = 13.2, 3.2 Hz, H-3’_b_), 1.25 (1H, m, H-5’_b_), 1.35 (1H, m, H-6’_b_), 1.69 (1H, t, J = 4.4 Hz, H-4’), 1.76 (1H, m, H-5’_a_), 2.03 (1H, m, H-6’_a_), 2.40 (1H, m, H-3’_a_), 4.99 (1H, d, *J* = 6.0 Hz, H-2’), 6.31 (1H, d, *J* = 16.0 Hz, H-3), 6.87 (2H, d, *J* = 8.4 Hz, H-6, H-8), 7.39 (2H, d, *J* = 8.4 Hz, H-5, H-9), 7.61 (1H, d, *J* = 16.0 Hz, H-2); ^13^C-NMR (CDCl_3_) δ: 13.5 (C-10’), 18.8 (C-9’), 19.6 (C-8’), 27.1 (C-6’), 30.0 (C-5’), 36.8 (C-3’), 44.8 (C-4’), 47.8 (C-7’), 48.9 (C-1’), 80.4 (C-2’), 115.2 (C-3), 115.9 (C-6, C-8), 126.5 (C-4), 130.0 (C-5, C-9), 144.0 (C-2), 158.6 (C-7), 168.0 (C-1); IR (KBr) ν_max_: 3343, 1682, 1517, 1171 cm^-1^; EI-MS (70 eV) *m/z* (rel. int.) 300 [M]^+^ (8), 164 (4), 147 (100), 136 (3), 119 (9), 109 (7), 95 (10), 91 (6), 81 (7), 69(8), 55 (6). The chemical structure of (+)-bornyl *p*-coumarate is shown in Figure 1.

### 2.2. (+)-Bornyl p-Coumarate Inhibits Melanocyte Cancer Cell Proliferation 

In this study, an MTT assay was employed to examine the efficacy of (+)-bornyl *p*-coumarate in terms of inhibiting melanocyte cancer cells. The results showed that (+)-bornyl *p*-coumarate was effective in two human melanoma skin cancer cell lines. With regards to an anti-proliferative effect, the results (Figure 2) showed that (+)-bornyl *p*-coumarate inhibited cell proliferation in A375 and A2058 cells, with an approximate 40% inhibition at 24 µM. We then explored whether (+)-bornyl *p*-coumarate affects apoptosis and autophagy in A2058 and A375 melanoma cells. Cells were treated with (+)-bornyl *p*-coumarate at 6~24 µM in the subsequent experiments so as not to affect the 50% cell survival rate. 

### 2.3. (+)-Bornyl p-Coumarate Induced Apoptosis in Melanoma Cells

As shown in Figure 2, (+)-bornyl *p*-coumarate inhibited the proliferation of A375 and A2058 cells. We then employed fluorescein isothiocyanate (FITC)-labelled annexin with propidium iodide (PI) to investigate the apoptotic pathways involved using flow cytometry in (+)-bornyl *p*-coumarate-treated A375 and A2058 melanoma cells. As shown in Figure 3A, in comparison with the control (0.1%), the percentages of late apoptotic cells in cells treated with 6, 12 and 24 μM (+)-bornyl *p*-coumarate were 14.9%, 15.6% and 17.3% in A2058 cells and 15.8%, 18.6% and 21.1% in A375 cells, respectively. The results showed that (+)-bornyl *p*-coumarate induced early and late apoptotic events in both cell lines in a dose-dependent manner. In order to understand whether apoptosis is involved in the inhibition of cell proliferation in A375 and A2058 cells by (+)-bornyl *p*-coumarate, we utilized a fluorescent TUNEL/DAPI assay to analyze the nuclear DNA integrity. The results showed that the green fluorescent intensity was amplified with increasing concentrations of (+)-bornyl *p*-coumarate concentration (Figure 3B).

(+)-Bornyl *p*-coumarate induced late apoptosis (Figure 3) in A375 and A2058 cells and studies have shown that apoptosis may cause mitochondrial inactivation. When cells become stressed or injured, the intracellular constant will be disturbed, resulting in changes to the permeability of the mitochondria. In order to understand whether (+)-bornyl *p*-coumarate causes apoptosis in A375 and A2058 cells via loss of mitochondrial function, we used immunostaining to analyze the effects of different concentrations of (+)-bornyl *p*-coumarate (0, 6, 12, 18 and 24 µM). After treatment of A375 and A2058 cells with (+)-bornyl *p*-coumarate, the mitochondrial-related protein expressions of the cells were examined. Analysis of (+)-bornyl *p*-coumarate-treated cells using western blotting indicated that the expressions of Bax and Bad increased with increasing concentration of (+)-bornyl *p*-coumarate, while the expressions of Bcl-2, Bcl-x1, Mcl-1, p-Bad and cytochrome *C* (cytosolic) decreased (Figure 4). Therefore, the addition of (+)-bornyl *p*-coumarate to A375 and A2058 cells exerted pressure on the mitochondria, which in turn increased the expressions of Bad and Bax and decreased those of cytochrome *C* (cytosolic), Bcl-2, Bcl-xl, Mcl-1 and p-Bad, leading to loss of function in mitochondria, as shown in Figure 4A. We also performed western blotting to examine the effects of (+)-bornyl *p*-coumarate on the activation of caspase-9, caspase-3 and PARP-1 cleavage. The results demonstrated that (+)-bornyl *p*-coumarate treatment up-regulated the expression levels of cleavage-PARP-1, cleavage-caspase-3 and cleavage-caspase-9 and down-regulated the expression levels of pro-caspase-3 and pro-caspase-9. (+)-Bornyl *p*-coumarate induced mitochondrial dysfunction and increased the level of caspase activation, leading to apoptosis in A375 and A2058 cells. In order to examine the role of the caspase cascade in (+)-bornyl *p*-coumarate-induced apoptosis in A357 and A2058 cells, Z-DEVD-FMK (a caspase-3 inhibitor) and Z-VAD-FMK (a cell-permeable pan-caspase inhibitor) were employed. As per the results shown in Figure 4B, reduction of cell death was observed in the cells treated with either of these two inhibitors, indicating that both caspase-3 and caspase-9 are involved in (+)-bornyl *p*-coumarate-induced apoptosis in these two melanoma cell lines.

### 2.4. (+)-Bornyl p-Coumarate Treatment Induced the Endoplasmic Reticulum (ER) Stress Pathway

Previous studies have indicated that ER stress is involved in the mechanism of apoptosis. In this study, immunostaining analysis following treatment of melanoma cells with (+)-bornyl *p*-coumarate revealed increases in the cell concentrations and expressions of ER-related proteins, including GRP78. In addition, three ER sensors were analyzed by western blot—PERK, endoribonuclease inositol-requiring enzyme 1α (IRE1α) and caspase-12. The results showed that the expressions of IRE1α and caspase-12 were unchanged after (+)-bornyl *p*-coumarate treatment. Under ER stress, a 50 kDa cleaved fragment (p50 activates transcription factor (ATF6)) gradually increased from the 90 kDa protein (p90 ATF6), which translocates to the nucleus and operates as a transcription factor to promote the expression of GRP78. The results indicated increased expressions of GRP78, *p*-PERK, *p*-eIF2α and ATF6-f after (+)-bornyl *p*-coumarate treatment (Figure 5A). The expressions of ATF4 and CHOP also increased with increasing (+)-bornyl *p*-coumarate concentration, indicating that (+)-bornyl *p*-coumarate may also cause apoptosis through ER stress. We then employed salubrinal, a phosphatase (PP1) inhibitor that selectively inhibits eIF2α dephosphorylation, to further confirm our hypothesis. An MTT assay showed that the survival rate of cells treated with (+)-bornyl *p*-coumarate and salubrinal together was significantly higher than that of cells treated with (+)-bornyl *p*-coumarate alone (Figure 5B).

### 2.5. (+)-Bornyl p-Coumarate Triggers Autophagy in Melanoma Cells

We performed western blot analysis to examine the expressions of autophagy-related proteins and the results showed that (+)-bornyl *p*-coumarate induced autophagy in melanoma cells, upregulating the protein levels of Beclin-1, LC3-I, LC3-II, Atg5, Atg3 and p62. The augmentation of these protein expressions was found to be greater with increasing concentrations of (+)-bornyl *p*-coumarate (Figure 6). Additionally, when autophagic inhibitor 3-methyladenine (3-MA) was employed, suppression of LC3-I, LC3-II and p62 expressions was observed (Figure 6B). The results indicated a pro-apoptotic effect of (+)-bornyl *p*-coumarate and treatment of the cells caused severe impairment of cellular functions, provoking cellular stress and inducing autophagy and apoptosis.

## 3. Discussion

There are two main pathways of apoptosis—the extrinsic pathway and the intrinsic pathway. The former is also known as the death receptor pathway or alternatively the mitochondrial pathway [21]. Studies have shown that these two pathways are interrelated and interact with each other [22] and endogenous pathways mainly occur in the ER or mitochondria [23]. Bcl-2 family proteins, also known as anti-apoptotic molecules, may block the intrinsic apoptosis pathway and cause breakage of the outer mitochondrial membrane, a process that may obstruct activation of Bad and Bax [24,25,26]. Caspase-dependent apoptosis is a known type of programmed cell death that has frequently been studied [27]. Mitochondrial inactivation in apoptosis is highly-associated with Bcl-2 family proteins and the balance between Bcl-2 (anti-apoptotic) and Bax (pro-apoptotic) proteins and cytochrome *C* is gradually released from the mitochondria to the cytoplasm, leading to caspase-9 activation and further activation of caspase-3, ultimately resulting in poly(ADP-ribose) polymerase cleavage [28,29,30]. Some studies have reported the release of cytochrome *C* into the cytoplasm triggered by mitochondrial inactivation, which consequently leads to caspase-3 and caspase-9 activation and results in apoptosis [31,32]. The results of western blot analysis demonstrated that the expression of Bax was gradually increased, while the expressions of Bcl-2/Bcl-xL were inhibited. Bax was translocated to the outer membrane of the mitochondria, causing the release of cytochrome *C* and further resulting in activation of caspase-3 and caspase-9. According to the results shown in Figure 5, reductions in the expressions of Bcl-2, Bcl-xl and Mcl-1 (anti-apoptotic proteins) and increases in Bad and Bax (pro-apoptotic proteins) after treatment with (+)-bornyl *p*-coumarate, in addition to activation of caspase-3, caspase-9 and PARP-1 (Figure 6), suggested that mitochondrial dysfunction and the caspase activation cascade play key roles in apoptosis in A2058 and A375 cells induced by (+)-bornyl *p*-coumarate.

One of the main functions of the ER is quality control of protein synthesis and folding and the membrane of the ER also plays important roles in calcium storage and intracellular signaling. The ER regulates protein-folding homeostasis with an unfolded protein response to ensure an excellent protein-folding capacity. The unfolded protein response relies on three ER stress sensor proteins, IRE1α, PERK and ATF6. Under ER stress, IRE1α quickly undergoes homo-oligomerization and autophosphorylation. Autophagy is known to be associated with PERK signaling, which is crucial for cell survival [33]. Alternatively, cells may induce apoptosis via enhanced expressions of ATF4 and proapoptotic transcription factor CCAA/enhancer binding protein (C/EBP) homologous protein (CHOP) [34]. When an unfolded protein response occurs, transcription and synthesis of client proteins are reduced due to phosphorylation of eIF2α by *p*-PERK and other ER stress sensor proteins, provoking the expressions of ER chaperones [35]. Our western blotting results showed an increased level of GRP78 following (+)-bornyl *p*-coumarate treatment, which suggested that activation of GRP78 reduces ER stress and reinstates normal protein folding. Activation of eIF2α, ATF4 and CHOP by PERK is known to cause apoptosis. Our western blotting analysis demonstrated that increased levels of *p-*PERK, *p*-eIF2α, ATF4 and ATF6 supported the pathways causing CHOP activation and were important in (+)-bornyl *p*-coumarate-induced apoptosis. These results revealed the significant role of the PERK-eIF2α-ATF4-CHOP signaling pathway in the ER in terms of the apoptotic mechanism induced by (+)-bornyl *p*-coumarate (Figure 5A). We observed that salubrinal rescued cell survival in cells treated with (+)-bornyl *p*-coumarate alone (Figure 5B), which confirmed the hypothesis that (+)-bornyl *p*-coumarate initiated ER stress and consequently induced apoptosis in melanoma cells.

Tumor cells in the presence of stress, a lack of nutrients or treatment with some anticancer drugs will induce autophagy to avoid damage and protect the cells [36,37] and many studies have indicated that inhibition of the late stage of autophagy causes cumulative cell phagocytic vacuoles to enter the cytoplasm, resulting in the death of cancer cells via mechanisms such as apoptosis. Several studies have demonstrated a connection between autophagy and tumorigenesis, as many central genes that regulate autophagy, such as Beclin-1, death-associated protein kinase (DAPK) and phosphatase and tensin homolog (PTEN), are also tumor suppressor genes [10,11,15]. Beclin-1, a regulator of autophagy initiation, was the first tumor suppressor gene to be found in autophagy-related proteins in human cancers. A decreased gene expression level of Beclin-1 has been noted in many breast cancer cells [16]. Beclin-1 and LC3-II, located on the membrane of autophagosomes, have been found to be downregulated together with p62/SQSTM1 in autophagy [38]. A lower level of Beclin-1 is known to be associated with human cancer progress. For example, suppression of Beclin-1 expression may promote tumorigenesis in mice [39]. Beclin-1 has the BH3 motif required for binding to Bcl-2, Bcl-xL and Mcl-1. Bcl-2 has been shown to down-regulate the pro-autophagic ability of Beclin-1, in which Bcl-2 binds to Beclin-1 to prevent Beclin-1 from interacting with class III PI-3K and decreasing class III PI-3K activity, leading to inhibition of autophagy [40]. Previous studies have indicated that natural compounds extracted from plants cause cell death in many different cancer cells by inducing autophagy in cells [41,42,43]. In this study, we employed (+)-bornyl *p*-coumarate to treat melanoma cell lines A375 and A2058. According to the experimental results, the expressions of Beclin-1, LC3-I, LC3-II, Atg3, Atg5 and p62 increased with increasing (+)-bornyl *p*-coumarate concentration (Figure 6A). 

(+)-Bornyl *p*-coumarate inhibited Bcl-2, Bcl-xL and Mcl-1 expressions, possibly releasing the autophagic activity of Beclin-1. At the same time, downregulation of Bcl-2, Bcl-xL and Mcl-1 by (+)-bornyl *p*-coumarate may induce autophagy-dependent apoptosis in melanoma cells. To that end, the effect of (+)-bornyl *p*-coumarate on the expression of survivin was investigated. Moreover, the addition of autophagy inhibitor 3-MA reduced the (+)-bornyl *p*-coumarate-induced cytotoxicity and the amounts of LC3-I, LC3-II and p62 in A375 and A2058 cells, demonstrating that (+)-bornyl *p*-coumarate induced tumor-suppressive autophagy. In conclusion, our results demonstrated that dysfunction of the mitochondrial pathway, activation of caspase cascades and ER stress are the main elements in (+)-bornyl *p*-coumarate-induced apoptosis and autophagy in A2058 and A375 cells.

## 4. Materials and Methods 

### 4.1. General Instrumental Operation for the Isolation and Identification of Compounds

Optical rotations were measured on a JASCO DIP-180 digital polarimeter (JASCO, Tokyo, Japan). IR spectra were obtained on a Perkin-Elmer 983 G spectrophotometer (Perkin-Elmer, Waltham, MA, USA). ^1^H and ^13^C nuclear magnetic resonance (NMR) spectra were recorded in CDCl_3_ at room temperature on a Varian-Unity-Plus-400 spectrometer with residual solvent signals as internal references (Varian, CA, USA). Chemical shifts are presented as δ values and coupling constants (*J*) are given in Hertz (Hz). Electron impact mass spectrometry (EI-MS) were recorded on a JEOL SX-102A mass spectrometer (JEOL, Tokyo, Japan). Column chromatography (CC) was performed on silica gel (230–400 mesh; Merck, Darmstadt, Germany) and Thin layer chromatography (TLC) analysis was carried out on pre-coated silica gel plates (Kieselgel 60 F-254; Merck, Darmstadt, Germany). Semi-preparative HPLC was performed using a normal-phase column (Purospher STAR Si, 5 mm, 250 × 10 mm; Merck, Darmstadt, Germany) and a reverse-phase column (Hypersil Gold C18, 5 μm, 250 cm × 4.6 mm; Thermo Scientific, Waltham, MA, USA) on an LDC Analytical-III system (LDC Analytical, FL, USA).

### 4.2. Preparation of P. betle Stem Extract

Stems of *P. betle* were collected in Pingtung County, Taiwan, in July 2008, which had been cultivated by a local farmer and identified by Professor Sheng-Zehn Yang, Curator of the Herbarium, National Pingtung University of Science and Technology. A voucher specimen was kept in the laboratory of Professor Chi-I Chang, Department of Biological Science and Technology, National Pingtung University of Science and Technology, Pingtung, Taiwan. After air-drying, *P. betle* stems (3 kg) were mechanically milled to a fine powder and sieved through a #10 mesh. The sieved powder was mixed with methanol at a powder—solvent ratio of 1:5 and extracted at room temperature for seven days and the process was repeated three times. After filtration and solvent evaporation, the crude extract was suspended in water, then partitioned with H_2_O and EtOAc successively. The ethyl acetate fraction (65 g) was chromatographed over silica gel and eluted with *n*-hexane-EtOAc and EtOAc-MeOH mixtures to give 24 fractions. Fraction 11 (3.2 g) was further chromatographed on a silica gel column (2 × 45 cm), eluted with CH_2_Cl_2_-EtOAc (99:1 to 1:1) and resolved into seven fractions (each of about 600 mL), 11A–11G. Fraction 11D was subjected to semipreparative HPLC and eluted with CH_2_Cl_2_-EtOAc (30:1) to yield (+)-bornyl *p*-coumarate (26.2 mg). (+)-Bornyl p-coumarate was dissolved in DMSO at a concentration of 10 mM as a stock solution and then diluted with PBS containing 10% DMSO *v/v.*

### 4.3. Reagents

Dulbecco’s modified Eagle’s medium (DMEM) and fetal bovine serum (FBS) were obtained from Biowest (Nuaillé, France). Polyvinylidene difluoride (PVDF) membranes and goat anti-rabbit and horseradish peroxidase-conjugated immunoglobulin (Ig) G were obtained from Millipore (Bellerica, MA, USA). MTT, 3-MA, salubrinal, Z-DEVD-FMK, Z-VAD-FMK and β-actin antibody were purchased from Sigma (St. Louis, MO, USA). An annexin V-FITC/PI apoptosis detection kit was purchased from Pharmingen (San Diego, CA, USA). Enhanced chemiluminescence (ECL) reagents were purchased from Pierce Biotechnology (Rockford, IL, USA).

### 4.4. Cell Culture and Drug Treatment

Human melanoma A375 and A2058 cell lines were obtained from the Bioresource Collection and Research Center (Food Industry Research and Development Institute, Hsin Chu, Taiwan). Cells were grown in DMEM with 10% FBS and maintained in a cell culture incubator with 5% CO_2_ at 37 °C. Two μl of sample solution were placed in each well of a 96-well plate. For all in vitro experiments, the final concentration of DMSO was 0.1% *v/v* and the solubility of (+)-bornyl *p*-coumarate in the tissue culture media was good, with no precipitation or turbidity. Cells were treated with different concentrations of (+)-bornyl *p*-coumarate (6, 12, 18 and 24 μM) and harvested after 24 h of incubation and the same final concentration of DMSO (0.1% *v/v*) was also used to treat the cells in the control group. To determine whether (+)-bornyl *p*-coumarate induced cell cytotoxicity via ER stress or mitochondria transmembrane potential depletion, cells were pretreated with salubrinal (10 µM) or 3-MA (10 µM) for 1 h before (+)-bornyl *p*-coumarate treatment.

### 4.5. Cell Viability Assay

The cell viabilities of A375 and A2058 cells treated with (+)-bornyl *p*-coumarate were measured using an MTT assay. In a 96-well plate, 1 × 10^4^ cells were seeded in each well and incubated with different concentrations of (+)-bornyl *p*-coumarate. MTT assays were performed as a standard method. All experiments were repeated three times under the same conditions.

### 4.6. Flow Cytometric Assessment of Apoptosis

A375 and A2058 cells were cultured with (+)-bornyl *p*-coumarate for 12 h and single-cell suspensions were collected in order to detect apoptosis using an annexin V-FITC/PI kit according to the manufacturer’s instructions. A FACScan flow cytometer (Becton-Dickinson) was used to analyze the cells and the cell data were processed using Cell-Quest software (Becton-Dickinson). 

### 4.7. Immunofluorescence Microscopy

A DeadEnd Fluorometric transferase dUTP nick end labeling (TUNEL) kit and 4’-6-diamidino-2-phenylindole (DAPI) and were obtained from Promega (Madison, WI, USA). DAPI and TUNEL assays were performed according to the instructions of the manufacturer. A375 and A2058 cells (1 × 10^5^ cells/well, 12-well microplate) were plated in each well of the plate and treated with different concentrations of (+)-bornyl *p*-coumarate (0, 12 and 24 µM). The (+)-bornyl *p*-coumarate-treated cells and untreated cells were fixed with 4% paraformaldehyde (dissolved in PBS) before staining. Cells after DAPI and TUNEL staining were then photographed under a fluorescence microscope (Olympus IX71 CTS).

### 4.8. Western Blotting

Mitochondrial and cytosolic cytochrome *C* were separated using a cytochrome *C* releasing apoptosis assay kit (Biovision, Milpitas, CA, USA). A375 and A2058 cells were treated with different concentrations of (+)-bornyl *p*-coumarate (0, 6, 12, 18 and 24 µM) for 24 h and the protein lysates were subjected to western blot analysis. Protein samples were separated by 12.5% SDS-PAGE and transferred to polyvinylidene difluoride (PVDF) membranes (Millipore). The membranes were then incubated overnight with different primary antibodies at 4 °C, washed three times with PBST (10 mM NaH_2_PO_4_, 130 mM NaCl and 0.05% Tween-20) and incubated with a secondary antibody conjugated with horseradish peroxidase for 1 h. After washing, the immunoreactive proteins to the antibody were visualized using ECL reagents (Pierce Biotechnology). 

### 4.9. Statistical Analysis

Results were pooled from three independent experiments. Data from the MTT and cell migration assays, in addition to flow cytometric data, were analyzed using Student’s *t*-test (Sigma-Stat 2.0, San Rafael, CA, USA). A *p* value < 0.05 was considered statistically significant.

## 5. Conclusions

Our results indicated the existence of anti-tumor effects of and apoptosis induction by (+)-bornyl *p*-coumarate in A2058 and A375 melanoma cells. Western blot analysis confirmed changes in the expressions of proteins associated with cell apoptosis and autophagy. Overall, these results could provide valuable information for the development of (+)-bornyl *p*-coumarate as a novel chemotherapy drug against human melanoma.

## Figures and Tables

**Figure 1 ijms-21-03737-f001:**
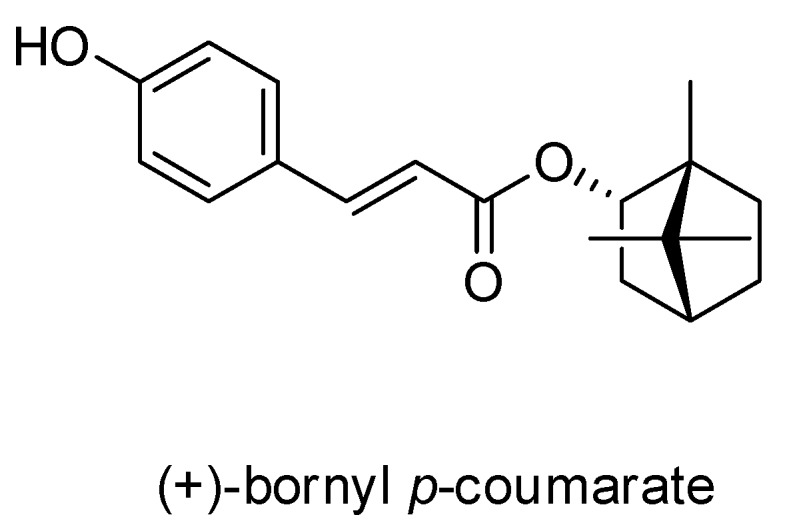
Chemical structure of (+)-bornyl *p*-coumarate.

**Figure 2 ijms-21-03737-f002:**
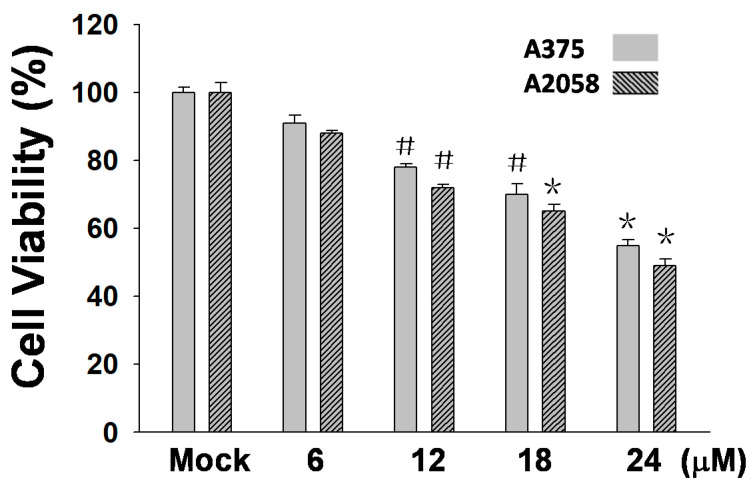
Results of an 3-(4,5-Dimethylthiazol-2-yl)-2,5- diphenyltetrazolium bromide (MTT) assay using varying concentrations of (+)-bornyl *p*-coumarate (0, 6, 12, 18, 24 µM). (+)-Bornyl *p*-coumarate inhibited melanocyte cancer cell proliferation in A375 and A2058 cells (^#^
*p* < 0.05; * *p* < 0.01). Mock: cells treated with vehicle control (DMSO).

**Figure 3 ijms-21-03737-f003:**
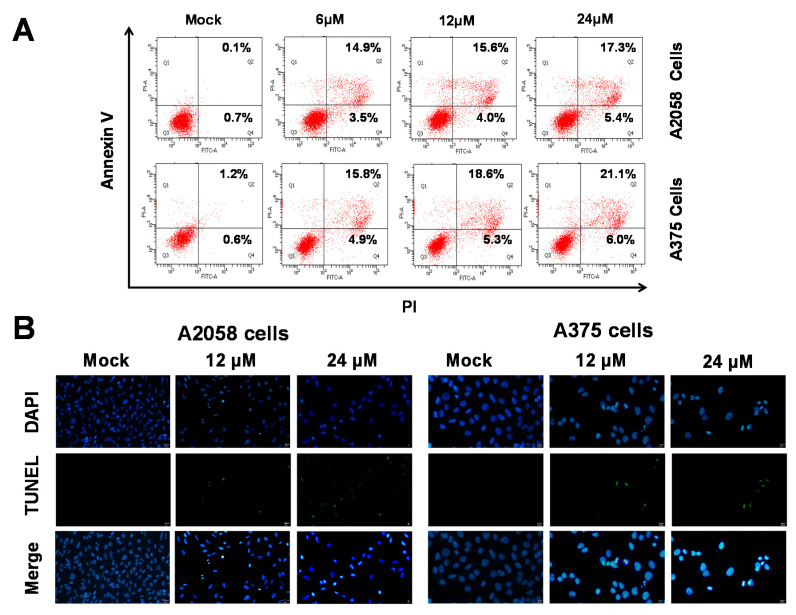
Detection of (+)-bornyl *p*-coumarate-induced apoptosis in A2058 and A375 melanoma cells by flow cytometry using FITC-annexin V and propidium iodide. (**A**) Percentages of late apoptotic cells in A2058 and A375 cells after (+)-bornyl *p*-coumarate treatment. (**B**) Apoptotic DNA fragmentation and nuclear condensation induced by (+)-bornyl *p*-coumarate, examined using a terminal deoxynucleotidyl transferase mediated dUTP nick end labeling (TUNEL) assay and 4,6-diamidino-2-phenylindole staining assay. Cells were treated with (+)-bornyl *p*-coumarate for 12 h at the indicated concentrations. Green fluorescence shows apoptotic DNA fragmentation, while blue DAPI staining indicates nuclear condensation. Mock: cells treated with vehicle control (DMSO).

**Figure 4 ijms-21-03737-f004:**
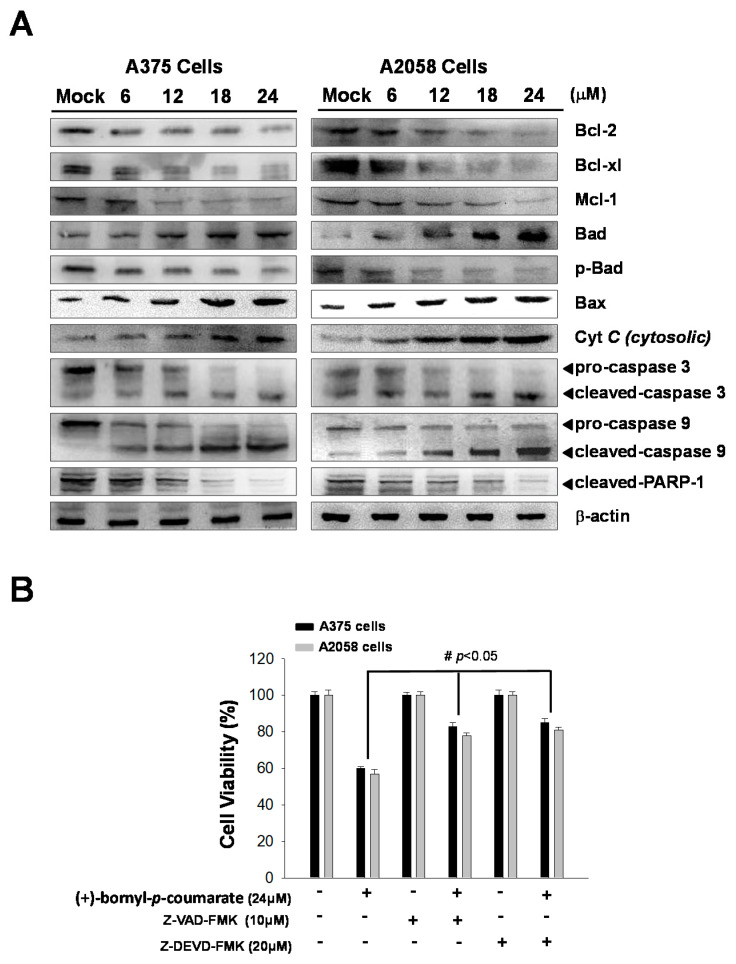
Mechanism of the caspase pathway in (+)-bornyl *p*-coumarate-induced apoptosis, validated by western blot analysis and an inhibitor assay. (**A**) Western blot analysis showed changes in apoptosis-associated proteins in A2058 and A375 melanoma cells after treatment with (+)-bornyl *p*-coumarate. (**B**) Inhibition of caspase-3 and caspase-9 suppressed (+)-bornyl *p*-coumarate-induced cell death. Cells were pre-treated with/without Z-DEVD-FMK and Z-VAD-FMK for 2 h, then treated with 12 µM (+)-bornyl *p*-coumarate for 24 h. Cell viability was measured using an MTT assay. Data are presented as the means ± SD of at least three independent experiments. The results were analyzed using Student’s *t*-test (^#^
*p* < 0.05 compared with (+)-bornyl *p*-coumarate treatment groups). Mock: cells treated with vehicle control (DMSO).

**Figure 5 ijms-21-03737-f005:**
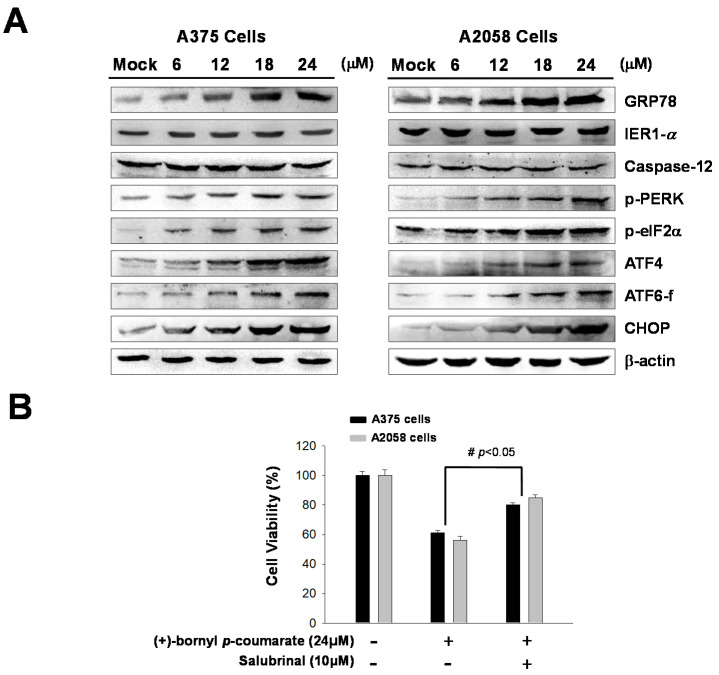
Analysis of the effects of (+)-bornyl *p*-coumarate on endoplasmic reticulum (ER) stress-associated proteins in A375 and A2058 cell lines. (**A**) Changes in ER stress response-related proteins in A2058 and A375 melanoma cells after (+)-bornyl *p*-coumarate treatment were evaluated by western blotting. (**B**) Incubation of cells with salubrinal demonstrated that the apoptosis caused by (+)-bornyl *p*-coumarate is mediated by the PERK–eIF2α–ATF4–CHOP pathway. Data are presented as the means ± SD of at least three independent experiments. The results were analyzed using Student’s *t*-test (^#^
*p* < 0.05, compared with (+)-bornyl *p*-coumarate treatment groups). Mock: cells treated with vehicle control (DMSO).

**Figure 6 ijms-21-03737-f006:**
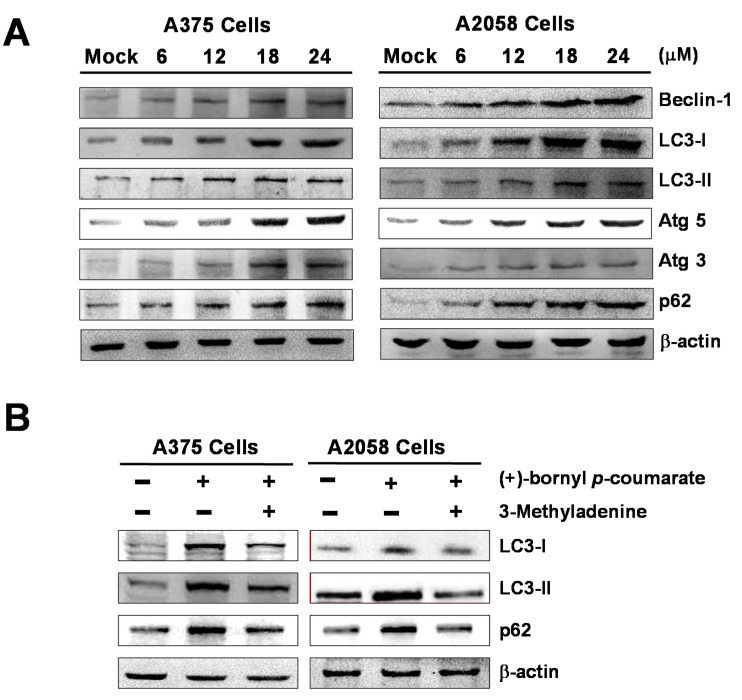
Induction of autophagy by (+)-bornyl *p*-coumarate. A357 and A2058 cells were treated with the indicated concentrations (6, 12, 18 and 24 μM) of (+)-bornyl *p*-coumarate for 24 h. (**A**) Autophagy-associated proteins, including Beclin-1, LC3-I, LC3-II, Atg3, Atg5 and p62, were analyzed using western blotting with antibodies. (**B**) The effects of 3-MA application (autophagic inhibitor; 20 μM) prior to treatment with 24 μM (+)-bornyl *p*-coumarate for 24 h on autophagy-associated proteins were also assessed. β-actin was used as the internal protein loading control. Mock: cells treated with vehicle control (DMSO).

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
