# Peer review of "(+)-Bornyl p-Coumarate Extracted from Stem of Piper betle Induced Apoptosis and Autophagy in Melanoma Cells"

_ijms, 2020, doi:10.3390/ijms21103737_

Round 1

Reviewer 1 Report

The authors have addressed all of my concerns.

Author Response

Thanks to reviewer for guidance.

Reviewer 2 Report

This research demonstrates that (+)-bornyl p-coumarate induces apoptosis and inhibits cell viability. The description of Figure 2 may be revised to add the reason why 50% cell survival rate was criteria. Careful proofreading including reference list is needed. It seems that reference 44 and 45 remain to be shown.

Author Response

Thank for reviewer’s suggestion. We have modified the description in figure 2 (line 104-108) and deleted reference 44 and 45.

Reply to the Reviewers’ Comments

Manuscript ID: IJMS-798465

Manuscript Title: (+)-Bornyl p-coumarate extracted from stem of Piper betle induced apoptosis and autophagy in melanoma cells

Authors : Yu-Jen Wu , Tzu-Rong Su, Chi-I Chang , Chiy-Rong Chen, Kuo-Feng Hung  and Cheng Liu

Dear Editor,

We would like to express our deep appreciation for your concern regarding the submitted manuscript. We are really thankful for your quick and informative reply. We are also grateful for the valuable comments and suggestions from the reviewers aiming to improve the scientific quality of the submitted manuscript. We replied to the comments and questions in a point-by-point fashion. Here we enclose the response to the reviewers’ comments.

We hope our response to the reviewers’ comments is satisfactory and meets IJMS standards. We also hope that our revised manuscript is acceptable for publication.

Thank you again for your time and concern.

Sincerely yours,

Professor Yu-Jen Wu, PhD

Department of Beauty Science,

Meiho University, Pingtung, TAIWAN

[email protected]; [email protected]

Reviewer 1

Comments and Suggestions for Authors

The authors have addressed all of my concerns.

Responds:

Thanks to reviewer for guidance.

Reviewer 3 Report

In this manuscript entitled "(+)-Bornyl p-coumarate extracted from stem of Piper 2 betle induced apoptosis and autophagy in melanoma cells" Yu-Jen et al showed that (+)-Bornyl p-coumarate is very effective in inhibiting melanoma cell proliferation, inducing apoptosis and autophagy.

This work cannot be published in this condition for the following reasons:

1. Extensive editing of English language and style required.

2. Additional experiments needed such as the effect of (+)-Bornyl p-coumarate on non-cancer cells and compare its impact with the cancer cells.

3. Almost every single figure needs to be replaced with a better quality figure.  

Round 2

Reviewer 3 Report

I would like to thank the authors for addressing all the comments.

This manuscript is a resubmission of an earlier submission. The following is a list of the peer review reports and author responses from that submission.

Round 1

Reviewer 1 Report

In this manuscript, Yu-Jen Wu and colleagues present the characterization of the effect of (+)-Bornyl p-coumarate molecule isolated from the stem of Piper betle over cancer cell lines. The authors claim that this compound has an important cytotoxic effect over two cancer melanoma cell lines via the induction of two major cell death pathways: apoptosis and autophagy. Although the general conclusions may be correct, the lack of suitable controls and/or the lack of methodological explanations makes this work uncomplete.

Specific points:

  • What is MOCK? The authors do not explain sufficiently what MOCK represents in the material/method section or in the result section? On line 402-403 we can read “…and DMSO of the same final concentration (0.01% v/v) was used to treat the cells in the control group…” Is this MOCK? If yes, why the authors use DMSO in just one dilution 1/10000 since they use 4 different doses of (+)-Bornyl p-coumarate meaning 4 different dilutions. So for each dilution of there has to be a MOCK.
  • The authors explain that the active compound is eluted in dichloromethane ethylacetate does this mean that they solubilized it additionally in DMSO? Why DMSO is used as MOCK and not CH2Cl2-EtOAc plus DMSO?
  • Why the authors use these four (+)-bornyl p-coumarate concentrations (6, 12, 18 and 24 µM). What is the rationality behind this? Especially if we take into account that the effect on the cell viability which is at its best 50% and not 60% as stated in line 118? What happens to the other 50% of the cell population if you use more concentrated drug or for a longer period? The authors should perform more extensive pharmacological analysis by changing one variable (time or concentration) at a time and perform more detailed analysis of the cell viability using the relevant controls.
  • It seems that there is a regulation of the quantity of Bcl-2 family proteins and Beclin. Is this managed on the transcriptional or post-translational level? Q-RTPCR should be performed in order to respond to this question. Is it phenomenon dependent on the UPR?
  • How the UPR is activated by (+)-bornyl p-coumarate? Is it because of increased synthesis of misfolded proteins? What about ER calcium levels are they affected?
  • How the cytosolic cytochrome C was obtained. There is no data in the material methods section.
  • In the figure 3A PI and Annexin V channels do not correspond to the arrows outside the squares
  • In the figure 3B the DAPI channel and Merge channel seem exactly the same. Why is this?
  • In the discussion section the figure names do not correspond to the figures presented in the result section. For example there is no figure 7…
  • In figure 4A I suppose that the authors present the cleavage of full-length PARP and not cleaved PARP.
  • The experiments with salubrinal are not presented in the results section. This should be corrected. Salubrinal is not eIF2α inhibitor (line 328). It is a phosphatase (PP1) inhibitor that a selectively inhibits eIF2α dephosphorylation. The same is for zVAD which is not caspase-9 inhibitor ( line 173) it is pan caspase inhibitor.
  • Generally speaking the consistency of the text should be improved.

Author Response

Reviewer 1.

In this manuscript, Yu-Jen Wu and colleagues present the characterization of the effect of (+)-Bornyl p-coumarate molecule isolated from the stem of Piper betle over cancer cell lines. The authors claim that this compound has an important cytotoxic effect over two cancer melanoma cell lines via the induction of two major cell death pathways: apoptosis and autophagy. Although the general conclusions may be correct, the lack of suitable controls and/or the lack of methodological explanations makes this work uncomplete.

Specific points:

  • What is MOCK? The authors do not explain sufficiently what MOCK represents in the material/method section or in the result section? On line 402-403 we can read “…and DMSO of the same final concentration (0.01% v/v) was used to treat the cells in the control group…” Is this MOCK? If yes, why the authors use DMSO in just one dilution 1/10000 since they use 4 different doses of (+)-Bornyl p-coumarate meaning 4 different dilutions. So for each dilution of there has to be a MOCK.

Responds:

Thank for reviewer’s suggestion. We have clearly defined the Mock (Mock: cells treated with vehicle control (DMSO), and put this statement in figure 1-6. We have modified this error and replaced the description in material and method (line 438-444).

  •  
  • The authors explain that the active compound is eluted in dichloromethane ethylacetate does this mean that they solubilized it additionally in DMSO? Why DMSO is used as MOCK and not CH2Cl2-EtOAc plus DMSO?

Responds:

Thank for reviewer’s suggestion. (+)-Bornyl p-coumarate is a dry powder after purification, in the experiment we used DMSO to dissolve (+)-Bornyl p-coumarate for subsequent cell experiments. Because DMSO can dissolve well the most of compounds and also it is not evaporating fast, we chose DMSO is used as MOCK and not CH2Cl2-EtOAc plus DMSO in the research.

  • Why the authors use these four (+)-bornyl p-coumarate concentrations (6, 12, 18 and 24 µM). What is the rationality behind this? Especially if we take into account that the effect on the cell viability which is at its best 50% and not 60% as stated in line 118? What happens to the other 50% of the cell population if you use more concentrated drug or for a longer period? The authors should perform more extensive pharmacological analysis by changing one variable (time or concentration) at a time and perform more detailed analysis of the cell viability using the relevant controls.

Responds:

Thank for reviewer’s suggestion. We used different concentrations (+)-bornyl p-coumarate (6 ~ 48 µM) when performing cell viability assay, only the results of (6 ~ 24 µM) were presented in figure 1. Because the higher the concentration, the lower the cell survival rate. In this study, we want to explore whether (+) -bornyl p-coumarate has the effect of apoptosis and autophagy on A2058 and A375 melanoma cell. We have used (+)-bornyl p-coumarate (24 µM) treatment cells in the experiment not to affect the 50% cell survival rate. The reviewer recommends that a more extensive pharmacological analysis should be performed by changing one variable (time or concentration) at a time, and a more detailed analysis of cell viability should be performed using relevant controls. I will study in this direction in future research.

  • It seems that there is a regulation of the quantity of Bcl-2 family proteins and Beclin. Is this managed on the transcriptional or post-translational level? Q-RTPCR should be performed in order to respond to this question. Is it phenomenon dependent on the UPR?

Responds:

Thank for reviewer’s suggestion. Because the research instruments and equipment in our research lab are not very perfect, we cannot perform on Q-RTPCR in our laboratory. In the future, we will collaborate with other research laboratories, and the reviewer recommends that it be performed in future experiments.

  • How the UPR is activated by (+)-bornyl p-coumarate? Is it because of increased synthesis of misfolded proteins? What about ER calcium levels are they affected?

Responds:

Thank for reviewer’s suggestion. In this study, we found (+)-bornyl p-coumarate increased synthesis of misfolded proteins, so it is speculated that (+)-bornyl p-coumarate activates UPR by increasing synthesis of misfolded proteins and activating PERK–eIF2α–ATF4–CHOP signaling pathway. Since we have not tested the ER calcium levels, it is impossible to make an absolute conclusion about the effect on ER calcium levels.

  • How the cytosolic cytochrome C was obtained. There is no data in the material methods section.

Responds:

Thank for reviewer’s suggestion. We have added the description in Materials and Methods 4.8 Western blotting. (line 471-473 )

  • In the figure 3B the DAPI channel and Merge channel seem exactly the same. Why is this?

Responds: Thank for reviewer’s suggestion. It’s our mistate. In the fluorescent photo section, we readjusted its fluorescent brightness. In addition, we processed 24 mM (+)-bornyl p-coumarate treatment fluorescent images in A2058 cells and made new results. And then we replaced the data in Figure 3B.

  • In the discussion section the figure names do not correspond to the figures presented in the result section. For example there is no figure 7…

Responds:

Thank for reviewer’s suggestion. We have modified this error.

  • In figure 4A I suppose that the authors present the cleavage of full-length PARP and not cleaved PARP.

Responds:

Thank for reviewer’s suggestion. We have modified this error and replaced in Figure 4A.

  • The experiments with salubrinal are not presented in the results section. This should be corrected. Salubrinal is not eIF2α inhibitor (line 328). It is a phosphatase (PP1) inhibitor that a selectively inhibits eIF2α dephosphorylation. The same is for zVAD which is not caspase-9 inhibitor ( line 173) it is pan caspase inhibitor.

Responds: Thank for reviewer’s suggestion. We have modified this mistake and restated in Result and discussion. (line 258-262)

  • Generally speaking the consistency of the text should be improved.

Responds: Thank for reviewer’s suggestion. We will continue to improve.

Reviewer 2 Report

The manuscript assesses the impact that (+)-bornyl p-coumarate (BPC) has on two melanoma cell lines. It further seeks to demonstrate the mechanism that this compound stimulates to induce cell death. While the authors clearly track the mechanism of death to an autophagic response induced through the ER stress pathway, there are a couple of critical points that must be addressed.

1) Figure 2 and later figures only demonstrate 40% suppression of cell growth following treatment with 24 uM doses of BPC. This must be correctly presented in the text where it is stated as 60% cell killing. Further, there is no discussion of the solubility of BPC in the tissue culture media, nor how the compound is introduced. The authors need to define what solution BPC is resuspended in and what resulting concentration of organic solvent is present in the treated cell lines. Further, that the mock contains this concentration of organic solvent. 

2) Figure 3 depicts cell killing following 12 hours of treatment. This is a very short time period and raises the concern that the compound at these high concentrations is simply insoluble and therefore is causing acute membrane damage/stress which is ultimately responsible for the ER stress readout that the authors are observing. The BPC solubility studies will help to address this, but the authors should also look at a later time-point (suggested 48 hours) for evidence of continued cell death and possibly apoptosis.

3) Figure 3: What is the timeframe for tunel assay staining? 24 hours as in the WB data below or 12 hours as used for the flow studies? Further, there is no discussion of the tunel assay in the text and this should be included.

4) Figures 4, 5, and 6: Please state what the mock treatment is for each of these western blots. 

Author Response

Reviewer 2

The manuscript assesses the impact that (+)-bornyl p-coumarate (BPC) has on two melanoma cell lines. It further seeks to demonstrate the mechanism that this compound stimulates to induce cell death. While the authors clearly track the mechanism of death to an autophagic response induced through the ER stress pathway, there are a couple of critical points that must be addressed.

  • Figure 2 and later figures only demonstrate 40% suppression of cell growth following treatment with 24 uM doses of BPC. This must be correctly presented in the text where it is stated as 60% cell killing. Further, there is no discussion of the solubility of BPC in the tissue culture media, nor how the compound is introduced. The authors need to define what solution BPC is resuspended in and what resulting concentration of organic solvent is present in the treated cell lines. Further, that the mock contains this concentration of organic solvent.

Responds:

Thank for reviewer’s suggestion. The (+)-bornyl p-coumarate was dissolved in DMSO as a stock solution and then it was diluted with PBS, containing 1% DMSO in v/v. A 2 ul of sample solution was introduced to each well of a 96-well plate. For all in vitro experiments, the final concentration of DMSO was 0.01% v/v, and the solubility of(+)-bornyl p-coumarate in the tissue culture media was good, no precipitation or turbidity was found. The same final concentration of DMSO (0.01% v/v) was also used to treat the cells in the control group. The above descriptions were added into the Materials and Methods section in Lines 438 and 444.

2) Figure 3 depicts cell killing following 12 hours of treatment. This is a very short time period and raises the concern that the compound at these high concentrations is simply insoluble and therefore is causing acute membrane damage/stress which is ultimately responsible for the ER stress readout that the authors are observing. The BPC solubility studies will help to address this, but the authors should also look at a later time-point (suggested 48 hours) for evidence of continued cell death and possibly apoptosis.

Responds:

Thank for reviewer’s suggestion.Both Flow cytometric assay and Immunofluorescence microscopy assay are tested by (+)-bornyl p-coumarate treated cells for 12 hr. The main purpose is to avoid the occurrence of cell necrosis when the cells are treated for too long. According to the reviewer's suggestion, prolong time the (+)-bornyl p-coumarate treatment cell to observe and search for evidence of continuous cell death and possible apoptosis, which will be designed in future experiments.

3) Figure 3: What is the timeframe for tunel assay staining? 24 hours as in the WB data below or 12 hours as used for the flow studies? Further, there is no discussion of the tunel assay in the text and this should be included.

Responds:

Thank for reviewer’s suggestion. Both Flow cytometric assay and Immunofluorescence microscopy assay are tested by (+)-bornyl p-coumarate treated cells for 12 hr. A375 and A2058 cells cells were treated with different concentrations of (+)-bornyl-p-coumarate (0, 6, 12, 18 and 24 µM) for 24 hrs. The protein lysates were subjected to western blotting analysis. We have added the description in Materials and Methods 4.8 Western blotting.

We have added the description about TUNEL/DAPI results in line 149-152.

4) Figures 4, 5, and 6: Please state what the mock treatment is for each of these western blots.

Responds:

Thank for reviewer’s suggestion. We have clearly defined the Mock (Mock: cells treated with vehicle control (DMSO), and put this statement in figure 1-6.

Round 2

Reviewer 2 Report

The authors performed some textual changes but failed to address 2 key points raised in the initial review.

1st- The statement that the compound induces 60% inhibition is false per the chart shown in figure 1. The maximum reduction in viability plotted is ~40%. This must be corrected in the text and a statement made as to why 24 uM was the highest dose tested.

2nd - The authors did not provide any statement on the aqueous solubility of the investigated compound. This is a critical point given the high concentrations used throughout the study.

These critical points need to be addressed to justify the remainder of the analyses that have been well conducted.